# Genomic Characterization of a Plasmid-Free and Highly Drug-Resistant *Salmonella enterica* Serovar Indiana Isolate in China

**DOI:** 10.3390/vetsci11010046

**Published:** 2024-01-20

**Authors:** Jiansen Gong, Ximin Zeng, Jingxiao Xu, Di Zhang, Xinhong Dou, Jun Lin, Chengming Wang

**Affiliations:** 1Poultry Institute, Chinese Academy of Agricultural Sciences, Yangzhou 225125, China; jjsensen@163.com (J.G.); shawn_zd@126.com (D.Z.); douxinhong611@163.com (X.D.); 2Jiangsu Co-Innovation Center for the Prevention and Control of Important Animal Infectious Disease and Zoonose, Yangzhou University, Yangzhou 225009, China; 3Department of Animal Science, The University of Tennessee, Knoxville, TN 37996, USA; xzeng3@utk.edu (X.Z.); jlin6@utk.edu (J.L.); 4School of Life Sciences, Fudan University, Shanghai 200438, China; caasxjx@sina.com; 5Department of Pathobiology, College of Veterinary Medicine, Auburn University, Auburn, AL 36849, USA

**Keywords:** highly drug-resistant *Salmonella enterica* serovar Indiana, ST17, resistance mechanism, genomic island, YeeVU toxin–antitoxin system

## Abstract

**Simple Summary:**

Researchers in this study were looking at certain types of *Salmonella* bacteria from China that are resistant to many drugs, and have found something surprising. They discovered a new type of this bacteria called S1467 that does not have the usual elements that make it resistant. When they studied S1467’s entire genetic makeup, they found that it has 54 genes that help it resist many different drugs. This shows how important it is to understand this bacterium better, especially how it becomes resistant to drugs. Furthermore, inside this S1467 bacterium, they found a unique section of DNA (about 51 kilobytes in size) that helps it resist drugs. This section also contains a special system called YeeVU, which is rare in this type of *Salmonella* bacteria. This new finding about S1467, a drug-resistant *Salmonella* type without the usual drug-resistance elements, is a big deal. It helps us understand more about the different types of *Salmonella* and how they become resistant to drugs. Knowing this helps scientists come up with better ways to fight *Salmonella* infections.

**Abstract:**

The emergence of multi-drug resistant (MDR) *Salmonella enterica* serovar Indiana (*S.* Indiana) strains in China is commonly associated with the presence of one or more resistance plasmids harboring integrons pivotal in acquiring antimicrobial resistance (AMR). This study aims to elucidate the genetic makeup of this plasmid-free, highly drug-resistant *S*. Indiana S1467 strain. Genomic sequencing was performed using Illumina HiSeq 2500 sequencer and PacBio RS II System. Prodigal software predicted putative protein-coding sequences while BLASTP analysis was conducted. The S1467 genome comprises a circular 4,998,300 bp chromosome with an average GC content of 51.81%, encompassing 4709 open reading frames (ORFs). Fifty-four AMR genes were identified, conferring resistance across 16 AMR categories, aligning closely with the strain’s antibiotic susceptibility profile. Genomic island prediction unveiled an approximately 51 kb genomic island housing a unique YeeVU toxin–antitoxin system (TAS), a rarity in *Salmonella* species. This suggests that the AMR gene cluster on the S1467 genomic island may stem from the integration of plasmids originating from other Enterobacteriaceae. This study contributes not only to the understanding of the genomic characteristics of a plasmid-free, highly drug-resistant *S*. Indiana strain but also sheds light on the intricate mechanisms underlying antimicrobial resistance. The implications of our findings extend to the broader context of horizontal gene transfer between bacterial species, emphasizing the need for continued surveillance and research to address the evolving challenges posed by drug-resistant pathogens.

## 1. Introduction

Non-typhoidal *Salmonella enterica* stands as a major global foodborne pathogen, causing approximately 78 million illnesses and 59,000 deaths annually [1,2,3]. Among the plethora of recognized *Salmonella* serovars, such as *Salmonella* Enteritidis and *Salmonella* Typhimurium [4,5], *Salmonella* Indiana (*S*. Indiana) has recently emerged as a notable player in the livestock and poultry industries in China, displaying an unusual rise in prevalence. Despite its global rarity, *S*. Indiana has become one of the most common serovars in China, with frequent reports of human infections [6]. Since 2008, *S*. Indiana has consistently ranked among the top three serovars in over 70% (46/65) of *Salmonella* isolations in China [6]. Notably, concomitant with this surge, there has been an alarming increase in highly drug-resistant *S*. Indiana strains [6,7,8,9].

Of particular concern is the observation that nearly 90% of *S*. Indiana isolates exhibit resistance to both fluoroquinolones and extended-spectrum cephalosporins, the primary therapeutic choices for severe salmonellosis [7,10,11]. Additionally, these strains manifest resistance to carbapenem and colistin, antibiotics designated as last-resort options for the treatment of multi-drug resistant (MDR) infections [12,13,14,15]. This underscores the gravity of the situation, signifying the emergence of MDR *S*. Indiana as a formidable public health issue.

The predominant multilocus sequence typing (MLST) type ST17, identified in 87.3% to 100% of the emerging Chinese *S*. Indiana isolates, has emerged as a leading epidemiologically significant type due to its heightened resistance to commonly used antibiotics [16]. Typically, ST17 *S*. Indiana strains harbor one or more resistance plasmids, notably a large plasmid, primarily of the IncHI2 type [12,14,17,18]. These plasmids are recognized for containing class I integrons that play a pivotal role in acquiring antimicrobial resistance (AMR) [19,20,21,22]. Intriguingly, a previous study revealed a unique plasmid-free *S*. Indiana strain, S1467 (ST17), in which the class I integron was identified within the chromosome [17]. In light of these findings, this study aims to provide a comprehensive understanding of the genetic characteristics of *S*. Indiana S1467 at the whole-genome level.

## 2. Materials and Methods

### 2.1. Strain Collection and Antimicrobial Susceptibility Testing

*Salmonella* Indiana S1467 was isolated from a broiler chicken fecal sample from Xuzhou, Jiangsu province, China [17]. The disk diffusion method was used to measure the zone diameter of streptomycin and the broth dilution method was used to determine the minimum inhibitory concentration (MIC) of the other 19 antibiotics (ampicillin, cefazolin, cefotaxime, ceftriaxone, ceftazidime, cefoxitin, aztreonam, imipenem, gentamicin, amikacin, kanamycin, tetracycline, nalidixic acid, ciprofloxacin, sulfonamides, trimethoprim, chloramphenicol, nitrofurantoin and colistin). The zone diameter values or MIC were determined in accordance with the methods of the Clinical Laboratory Standards Committee (CLSI). *Escherichia coli* ATCC25922 was used as the quality control strain [23].

### 2.2. Whole Genome Sequencing, Assembly and Annotation

Genomic DNA was extracted from *S*. Indiana S1467 using the bacterial genomic DNA extraction Kit (Tiangen Biotech Co., Ltd., Beijing, China). The genome was sequenced using a combination of Illumina HiSeq 2500 sequencer and the PacBio RS II System, at the Shanghai Biotechnology Corporation (Shanghai, China). A total of 5 µg of *Salmonella* genomic DNA was fragmented for PacBio library construction using the DNA Template Prep Kit 2.0 (Pacific Biosciences, Menlo Park, CA, USA). This was followed by sequencing and assembly using SPAdes genome assembler (v3.11.1) and A5-miseq (v20160825). AMPure magnetic beads (Beckman Coulter, Indianapolis, IN, USA) was used to purify the 8–12 kb fragments. Total length of 88,080 reads were obtained, with mean read length and mean subread length 23,375 bp and 3223 bp, respectively. The average coverage is 421.

Putative protein-coding sequences (CDSs) were predicted by the Prodigal_v2.6.1 software. The CDS annotation was based on the BLASTP program, with NR (http://ftp.ncbi.nih.gov/blast/db/, accessed on 20 September 2023), COG (http://eggnogdb.embl.de/#/app/home/), KEGG (http://www.genome.jp/kegg/), GO (http://www.geneontology.org/) and Swissprot (http://www.uniprot.org/) databases. The whole genome of *S*. Indiana S1467 has been deposited in the GenBank database, under the accession number CP121189. The sequencing data generated from this study have been deposited in NCBI under BioProject PRJNA949292.

### 2.3. Sequence Analysis of the S. Indiana S1467 Genome

The genome of *S*. Indiana S1467 was searched for AMR genes, predicted using the Comprehensive Antibiotic Resistance Database (CARD) (https://card.mcmaster.ca/). The genome was searched for CRISPR loci using CRISPR finder (http://crispr.i2bc.paris-saclay.fr/Server/). *Salmonella* pathogenicity islands (SPIs) and genomic islands in the *S*. Indiana S1467 genome were searched using SPIFinder 2.0 (https://cge.food.dtu.dk/services/SPIFinder/) and IslandViewer 4 (http://www.pathogenomics.sfu.ca/islandviewer/), respectively. We utilized the OrthoFinder software (V2.5.4) to perform ortholog analysis on the assembled genomes sourced from NCBI, specifically targeting single-copy genes. For phylogenetic analysis of the genome, we employed the MAFFT software (V7.271). Subsequently, a phylogenetic tree was generated using the ML algorithm in FastTree software (V2.1.11), based on sequences of single-copy genes. The resulting tree was then visualized using FigTree. (http://tree.bio.ed.ac.uk/software/figtree/). Easyfig was used to perform collinearity analysis between the genomic island and whole genome sequences of *Salmonella enterica*, *Klebsiella pneumoniae* and *Escherichia coli*.

## 3. Results

### 3.1. General Features of the S. Indiana S1467 Genome

The complete genome of *S*. Indiana S1467 contained a circular 4,998,300 bp chromosome (Figure 1), with an average GC content of 51.81%. It was predicted to encode 4709 open reading frames (ORFs), 84 tRNAs, 22 rRNAs and three CRISPRs. Alignment to the NR, COG, KEGG, GO and Swissprot databases resulted in 4694, 4355, 3061, 3719 and 4049 annotated genes, respectively. It was found that *S*. Indiana S1467 possessed six SPIs (SPI-1, SPI-2, SPI-3, SPI4, SPI-5 and SPI-9) and a lot of genomic islands (Appendix A). Based on the Blastn results of a 51 kb AMR genomic island analysis from the NCBI database, we selected *Salmonella enterica* (six strains), *Klebsiella pneumoniae* (five strains) and *Escherichia coli* (six strains) with a high similarity to the *S*. Indiana S1467 genomic island sequence. The ortholog analysis of the *S*. Indiana S1467 genome, with another 15 Enterobacteriaceae genomes, is shown in Figure 2. There were 2583 clusters that were shared by all selected bacteria, while the total number of orthologous clusters from S1467 was 4393 and the number of singletons was eight. The phylogenetic tree, based on single-copy genes (Figure 3), grouped 16 Enterobacteriaceae bacteria into three clusters based on different genera (*Salmonella enterica*, *Klebsiella pneumoniae*, and *Escherichia coli*), with *S*. Indiana S1467 exhibiting the highest homology to *S*. Indiana S530.

### 3.2. Antimicrobial Resistance in S. Indiana S1467

In this study, we found that *S*. Indiana S1467 was susceptible to three drugs (cefoxitin, imipenem and colistin) while exhibiting resistance to 17 other drugs spanning eight distinct antimicrobial classes (β-lactams, monobactams, aminoglycosides, tetracyclines, quinolones, folate pathway inhibitors, phenicols and nitroimidazoles) (Table 1). Consequently, *S*. Indiana S1467 presented an exceptionally resistant profile, showcasing resistance to at least eight different classes of drugs.

### 3.3. Antimicrobial Resistance Genes and Cassettes

The CARD database was queried to identify resistance-related genotypes in the genome of *S*. Indiana S1467. A total of 54 AMR genes were identified and encoded resistance to 16 different AMR categories (Table 1). In addition, the quinolone resistance determining region (QRDR) was identified. Double *gyrA* mutations (S83F and D87N) and a single *parC* mutation (S80R), which were located within the QRDR region, were detectable in S1467. Furthermore, we found three AMR gene cassettes; *mphE*-*msrE*+*armA*+*sul1*-*qacE*-*arr3*-*catB3*-*bla*_OXA1_-*aac(6′)-Ib*-*cr6*, *sul2*-*aph(3″)-Ib-aph(6)-Id+tet(A)* and *dfrA17*-*aadA5*+*oqxB*-*oqxA*, and multiple mobile elements, which included *ISCR1*, *IS26*, *IS6* and *IntI1*, upstream or downstream of the gene cassettes. These AMR genes and cassettes likely contribute to resistance against clinical drugs commonly used for salmonellosis treatment. Notably, the study found a consistent correlation between resistance phenotypes and AMR genotypes, with the absence of corresponding resistance genes for the three susceptible drugs (cefoxitin, imipenem, and colistin) in *S*. Indiana S1467.

### 3.4. Characteristics of a Drug-Resistance Genomic Island That Harbors YeeVU-TAS

Genomic island prediction revealed an approximately 51 kb (646,462 to 697,754 bp) AMR genomic island that contains a Type IV toxin antitoxin system (TAS), named YeeVU, two plasmid replication related genes that encode the replication initiation protein, *repM* [24], and the plasmid replication DNA-binding protein. Ten AMR genes, which included one of the above-mentioned AMR gene cassettes (*mphE*-*msrE*+*armA*+*sul1*-*qacE*-*arr3*-*catB3*-*bla*_OXA1_-*aac(6′)-Ib*-*cr6*), were also found on the genomic island. This genomic island mediates the resistance to β-lactams, aminoglycosides, folate pathway inhibitors, macrolides, phenicols, rifamycin and disinfecting agents drug classes. Based on the NCBI database, the Blastn results of the genomic island analysis showed that other Enterobacteriaceae also have drug-resistant genomic islands similar to that found in this study (Appendix A). We selected *Salmonella enterica* (six strains), *Klebsiella pneumoniae* (three strains) and *Escherichia coli* (five strains) with a high similarity for phylogenetic analysis. The results showed that *S*. Indiana S1467 is most closely related to *S*. Indiana YZ20MCS6 and *S*. Indiana SJTUF14152 (Figure 4). A collinearity analysis showed that the AMR genomic island had the greatest number of regions in common with *S*. Indiana YZ20MCS6 (Figure 5). Furthermore, the highly similar AMR gene cluster located on the *S*. Indiana S1467 genomic island was also found in other Enterobacteriaceae and was predominantly located on resistance plasmids.

## 4. Discussion

*Salmonella* serovar Indiana exhibits intricate mechanisms leading to elevated drug resistance, notably facilitated by highly transmissible elements such as integrons or conjugating plasmids. The heightened resistance, especially to first-line drugs like cephalosporins, poses a challenge in selecting suitable clinical medications for patients with *S*. Indiana infections. It is imperative to comprehensively investigate resistance genotypes and phenotypes across different sources of *S*. Indiana isolates. Understanding whether these resistance mechanisms can transfer between distinct *Salmonella* serovars and other pathogenic bacteria is crucial for evaluating health risks in community populations and hospital patients. Preventing and treating infections caused by clinically resistant bacteria necessitates a deeper understanding of these resistance mechanisms [25]. While there are numerous genomic characterizations of drug-resistant *S*. Indiana strains, the significance of *S*. Indiana S1467 lies in being the first reported plasmid-free, highly drug-resistant strain.

A low-quality draft genome with 467 contigs was generated in the previous study [17], while 24 AMR genes were identified in S1467. In comparison, the hybrid assembly of Illumina and PacBio sequencing was used to generate a high-quality complete genome with a total of 45 identified AMR genes. AMR genes often clustered in the genome, can be transmitted through mobile elements like plasmids, integrons, and transposons. Our study reveals a diverse and abundant presence of AMR genes on the chromosome of *S*. Indiana S1467, as identified through the CARD database. The correlation between AMR genotypes and phenotypes underscores the close association observed in our findings. Notably, the classical integron structure, comprising 5′ conserved segment (5′ CS), 3′ conserved segment (3′ CS), and a variable region (VR), was disrupted in *S*. Indiana S1467. The 5′ CS includes the *intI* gene, whilst the 3′ CS contains *qacE* and *sul1*. The VR harbors different acquired antibiotic resistance cassettes. In a previous study, we showed that *S*. Indiana S1467 carries a complete class I integron, which contains *qacE* and *sul1* in Contig 33 and the *intI* gene in Contig 108 [17]. In this study, we assembled the complete genome, using a combination of Illumina HiSeq 2500 sequencer and the PacBio RS II System, and found that the 5′ CS is distant from the VR and 3′ CS. This may be due to assembly errors caused by the repeated region (IS6) between the VR and 3′ CS. In addition, we found three CRISPRs in the S1467 genome and it has been suggested that the CRISPR-cas system may have the capacity to mobilize AMR genes from plasmids into the chromosome. This may contribute to the expansion of the drug resistance spectrum [26].

Unlike most MDR *S*. Indiana, *S*. Indiana S1467 does not bear plasmids that contain AMR genes. However, we found an abundance of AMR genes and three AMR gene cassettes located in the genome. Homology analysis of the AMR gene cluster located on the genomic island revealed that similar AMR gene cassettes are also present in other Enterobacteriaceae and are mainly located on resistance plasmids in *Escherichia* species, *Klebsiella* species, *Proteus* species and *Citrobacter* species. In addition, we found two plasmid replication-related genes downstream of this gene cluster. Therefore, we speculated that the resistance gene cluster on the genomic island of *S*. Indiana S1467 may have been transferred from plasmids of other Enterobacteriaceae species to *S*. Indiana S1467 chromosomes. Furthermore, this AMR gene cluster may have the potential for horizontal transfer into other Enterobacteriaceae.

We found a YeeVU-TAS on the genomic island of *S*. Indiana S1467. This was first reported in the *E. coli* genome and is a relatively uncommon type IV TAS [27]. Interestingly, we found four YeeVU-TAS in the S1467 genome and they were found in a repeat region. Their presence in this region may be due to recombination or plasmid integration events. Based on the NCBI database, we found that YeeVU-TAS is only distributed in Enterobacteriaceae such as *E. coli*, Shigella, *Klebsiella pneumoniae*, *Yersinia enterocolitica*, etc. [28]. As this TAS is rarely found in *Salmonella*, we speculated that YeeVU-TAS in *S*. Indiana S1467 originates from other Enterobacteriaceae through horizontal transfer or otherwise. The TAS consists of two co-expressed genes, one encoding a stable toxin (proteins) and the other encoding unstable antitoxins (proteins or RNA). The function of TAS is to maintain the stability of plasmids or genomic islands in bacteria. When plasmids (genomic islands) are lost in the offspring of bacteria, unstable antitoxin proteins will be rapidly degraded by proteases, producing free toxin proteins that kill the bacteria that have not retained plasmids (or have lost genomic islands) [29]. Research has also found that TAS interferes with or alters DNA replication, ATP and cell wall synthesis under environmental stimuli or stress. This leads to cell growth inhibition, death and the formation of persistent bacteria [30,31,32]. The biological significance of YeeVU-TAS in *S*. Indiana, involving two co-expressed genes encoding a stable toxin and unstable antitoxins, warrants further in-depth research. This TAS may play a role in maintaining plasmid or genomic island stability, impacting bacterial growth, persistence, and response to environmental stimuli or stress.

## 5. Conclusions

In this study, we used whole genome sequencing and assembly to generate the complete genome sequence for a plasmid-free, highly drug-resistant *S*. Indiana strain, S1467 (ST17), in which the class I integron is located on the chromosome. We found 54 AMR genes and three AMR gene cassettes in the *S*. Indiana S1467 genome. These mediate the resistance to many clinical drugs. Genomic island prediction revealed an approximately 51 kb AMR genomic island that contains a unique YeeVU-TAS that is rarely present in *Salmonella* species. This study contributes not only to the understanding of the genomic characteristics of a plasmid-free, highly drug-resistant *S*. Indiana strain but also sheds light on the intricate mechanisms underlying antimicrobial resistance. The implications of our findings extend to the broader context of horizontal gene transfer between bacterial species, emphasizing the need for continued surveillance and research to address the evolving challenges posed by drug-resistant pathogens.

## Figures and Tables

**Figure 1 vetsci-11-00046-f001:**
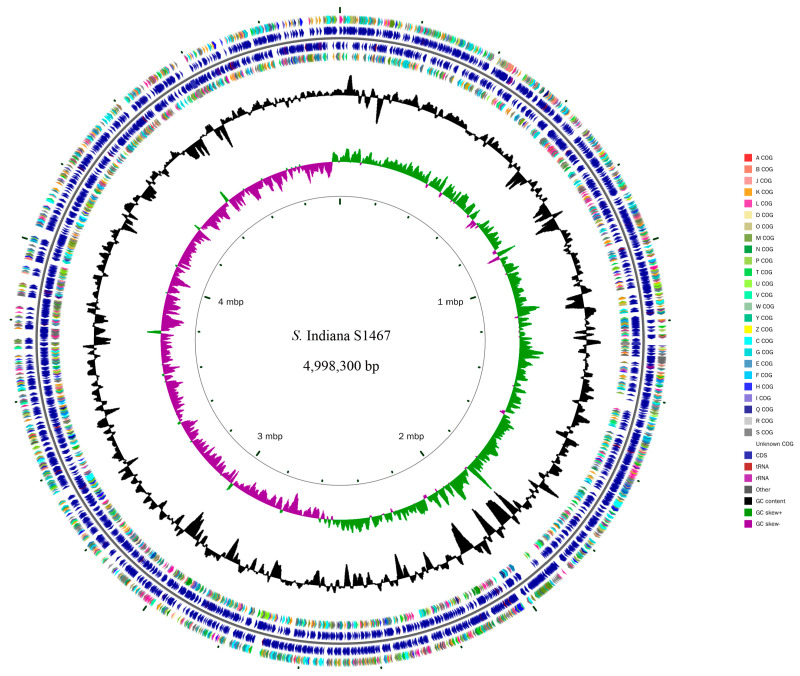
Circular visualization of the completed chromosome of *S*. Indiana S1467. The rings show (from inside to outside) the scale in mb, GC skew, GC content, CDS; position of CDS, tRNA, and rRNA.

**Figure 2 vetsci-11-00046-f002:**
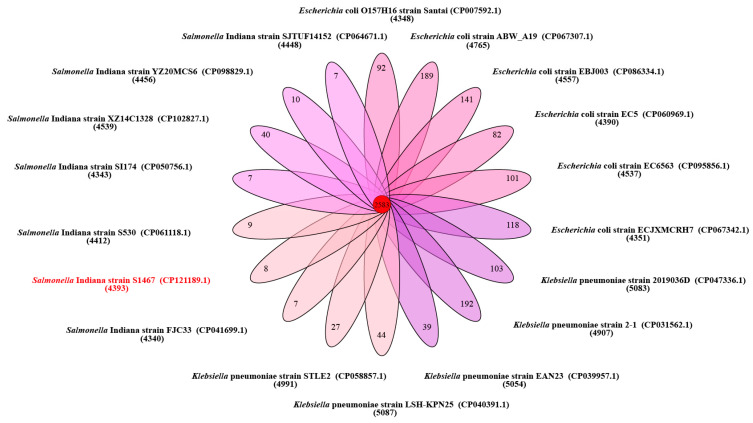
Gene orthology analysis of *S*. Indiana S1467 (text in red) and related Enterobacteriaceae strains. Venn diagram represents the numbers of unique and shared orthologous clusters between *S.* Indiana S1467 (text in red) and related Enterobacteriaceae strains.

**Figure 3 vetsci-11-00046-f003:**
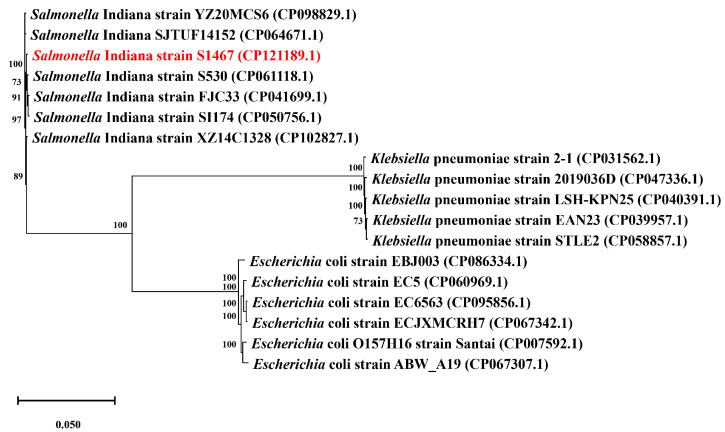
Concatenated core genome tree highlights the phylogenetic relationship between *S*. Indiana S1467 (shown in red) and other Enterobacteriaceae strains. The tree was constructed using 2583 core genes and is represented as a cladogram. The corresponding Genbank accession numbers are displayed in parentheses.

**Figure 4 vetsci-11-00046-f004:**
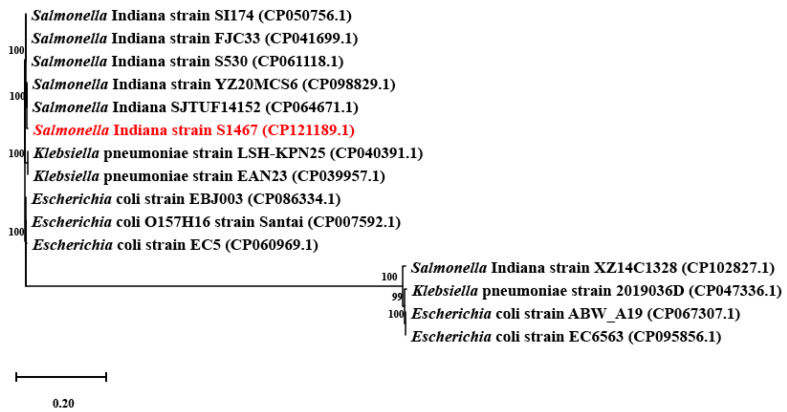
Concatenated core genome tree highlights the position of the *S*. Indiana S1467 (shown in red) AMR genomic island, relative to other Enterobacteriaceae strains. The corresponding Genbank accession numbers are displayed in parentheses.

**Figure 5 vetsci-11-00046-f005:**
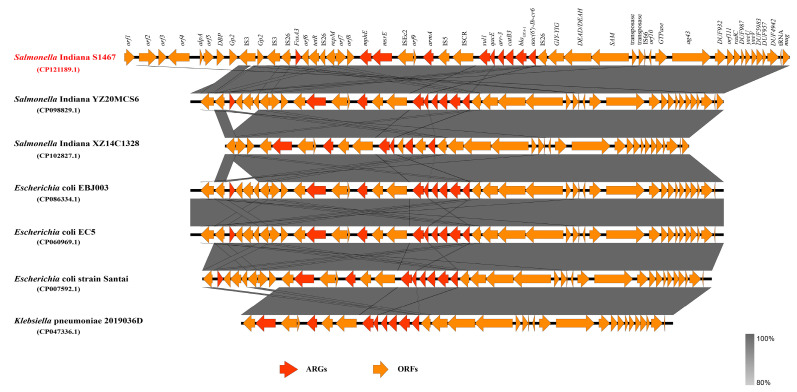
Major structural features of the S1467 AMR genomic island (shown in red), when compared with the corresponding chromosomal regions of other Enterobacteriaceae strains from the NCBI database. The arrows indicate the extent and direction of transcription of the genes. The open reading frames (ORFs) with different functions are presented in various colors (red: AMR genes, yellow: ORFs). The corresponding Genbank accession numbers are displayed in parentheses.

**Table 1 vetsci-11-00046-t001:** Antimicrobial resistance phenotypes and genes of *S*. Indiana S1467.

Classes	Resistance Phenotypes	Resistance Genes
β-lactams	ampicillin, cefazolin, cefotaxime, ceftriaxone, ceftazidime	*bla*_OXA-1_, *bla*_CTX-M-55_, *ftsI*
monobactams	aztreonam	*golS*, *mdsA*
aminoglicosides	gentamicin, amikacin, kanamycin, streptomycin	*armA*, *aph(3″)-Ib*, *aph(6)-Id*, *aadA5*, *aph(4)-Ia*, *aac(3)-Iva*, *aac(6′)-Iy*, *baeR*, *kdpE*
tetracyclines	tetracycline	*tetA*, *tetR*, *mdfA*
quinolones	nalidixic acid, ciprofloxacin	*rsmA*, *oqxB*, *oqxA*, *mdtK*, *sdiA*, *marA*, *acrA*, *acrB*, *adeF*, *soxR*, *soxS*, *acrAB-tolC*, *aac(6′)-Ib-cr6*
folate pathway inhibitors	sulfonamides, trimethoprim	*sul1*, *sul2* ^a^, *dfrA17*
phenicols	chloramphenicol	*catB3*, *floR* ^a^
nitroimidazoles	nitrofurantoin	*msbA*
rifamycin	ND ^b^	*arr3*
elfamycin	ND	*EF-Tu* ^a^
glycopeptides	ND	*vanG*
macrolides	ND	*CRP*, *mphE*, *msrE*, *emrB*, *emrR*, *H-NS*, *kpnF*, *kpnE*
nucleoside	ND	*leuO*
peptides	ND	*bacA*, *arnT*, *pmrF*
phosphonic acid	ND	*fosA3*, *glpT*
disinfecting agents and antiseptics	ND	*qacE*

^a^ Two copies were identified on the chromosome. ^b^ ND indicates that there was no drug tested for this category.

## Data Availability

The data contained within the article.

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
