# Peer review of "Genomic Characterization of a Plasmid-Free and Highly Drug-Resistant Salmonella enterica Serovar Indiana Isolate in China"

_vetsci, 2024, doi:10.3390/vetsci11010046_

Round 1
Reviewer 1 Report
Comments and Suggestions for Authors
Manuscript vetsci-2766385 describes the whole genome analysis of S. Indiana strain S1467. The article is well written; however, the present version suffers from several major limitations.
1. Simple Summary is too extensive and complex lacking the simplicity and clarity for the reader. As presented, seems a paraphrased version of the summary. It is recommended that the Simple Summary depicts the key findings of the research.
2. A big problem with the manuscript is that this version is incomplete and impossible to judge. The whole set of figures were not included in the document as indicated in Instructions for Authors.
3. As presented, the current version of the Brief Report is too descriptive. This version details the genomic finds after whole genome sequencing. Although this information could be relevant, the authors should compare these genomic features with other multiresistant isolates gains insights into mechanistic or evolutionary traits responsible for this pathogenic phenotype.
Author Response
Please refer to the PDF file in the attachment.

Reviewer 2 Report
Comments and Suggestions for Authors
Comments to the Author
In this manuscript, Gong et al. provide a comprehensive understanding of the genetic characteristics of a plasmid-free and highly drug-resistant S. Indiana S1467 at the whole-genome level. This study expands our understanding of the intricate antimicrobial resistance mechanisms of S. Indiana and horizontal gene transfer between bacterial species. In general, the manuscript is fairly well-written and presents interesting data. To improve the manuscript, following minor changes are suggested:
1) Lines 45, 110 and 220, the correct name is “Illumina HiSeq 2500 sequencer”;
2) Line 87, “class I integrons” should be more suitable;
3) Line 140, “six” should be changed to “eight”.
4) The accession number CP121189 for this isolate was provide.
5) Did the authors identify other close strains to this one? If yes, where were the origin of these additional strains? From what animal species? From environmental samples?
6) In this manuscript, the authors use the term AR while many others use AMR. The authors may consider switching to use AMR to replace AR?
In my opinion, the question addressed in this manuscript is tightly relevant to this Special Issue, and the findings are interesting. This serovar (Salmonella Indiana) is widely prevalent in China, and this strain used in this work is drug resistant but lacks plasmid. The conclusion from this work will surely help curb the spreading of S. Indiana in China and other countries. The main part of the originality of this work comes from the findings that this isolate lacks plasmid. While most other reported S. Indiana strains were reported to carry plasmid, this strain is totally different. The findings does help in the understanding of the pathogens’ diversity and, more importantly, the dynamics of antimicrobial resistance. It seems to me that the experiment and assays in this work are well described, and the conclusions are well supported by the assays. The authors of this manuscript are well known in the areas of foodborne pathogens and I trust the conclusion from this work. I enjoyed reading this manuscript, and this reviewer does not have any issues in understanding this manuscript.
Comments on the Quality of English Language
1) I suggest changing the number of AR genes from 57 to 54 (lines 27, 48, 161 and 265) and “two genes” changed to “two copies” (Table1 line303);
2) Line 31, “This island” should be changed to “This genome island”
Author Response

(The authors gave the same response as above.)

Reviewer 3 Report
Comments and Suggestions for Authors
The study is interesting due to high resistance genes found in this strain. However, the results are not properly presented. The authors should include more analysis, indicating the number of SPI present in the strains and indicate more clearly the genomic island carrying resistance genes that they refer.
Line 13: The authors include a simple summary section that I don´t see in the authors instructions section.
Line 79: was isolated instead of “was cultured”.
Line 91: The authors have to indicate the method used for genome assembly. This is important. Also the kits used for library preparation were not indicated. Number of reads obtained, the coverage….
Line 98: The accession to the raw reads?
Line 106: Version of MAFFT?
Line 108: Method used for tree construction.
Line 109: What genomes of Klebsiella pneumoniae and Escherichia coli were used?
Line 123: Section 3.2 should be section 3.1 and section 3.1 should be section 3.2.
Line 128: It is not clear what genomic island the authors refers to. There are many genomic island in Salmonella genome, so please indicate the specific genomic island.
Line 129: and 130: I don´t understand why Klebsiella pneumoniae and Escherichia coli.
Line 162: It should be interesting to add more information of those Salmonella Indiana that carry a similar plasmid, the origin for example, to understand the circulation of this plasmid. Are those Indiana, clinical strains? Also, by including the other species in the phylogenetic analysis, it is impossible to see the distances between the Salmonella Indiana.
Discussion section is to repetitive, please be concise.
Author Response

(The authors gave the same response as above.)

Reviewer 4 Report
Comments and Suggestions for Authors
This investigation employed whole genome sequencing and assembly techniques to decipher the entire genetic sequence of the S. Indiana strain, S1467. This strain is characterized by its lack of plasmids and high resistance to multiple drugs. The study identified 57 antimicrobial resistance (AR) genes, encompassing resistance against 16 distinct AR categories. Moreover, the prediction of a genomic island revealed an AR-associated region housing a Type IV toxin-antitoxin system (TAS) termed YeeVU. Although this study provides some meaningful data, several aspects require clarification and improvement.
1. Regarding section 3.1 detailing the Antimicrobial Resistance of S. Indiana S1467, reference 17 (PMID: 30866757) had previously conducted antimicrobial resistance analysis (seen in table 2 of ref. 17). Both studies demonstrate the susceptibility of S1467 to imipenem. A comparative statement is necessary to differentiate the findings between these two studies. For instance, while the previous study reported S1467's resistance to ..., the current study provides further insights into..., clearly indicating new discoveries versus established knowledge.
2. Regarding the Antimicrobial Resistance Genes and Cassettes section, this study identifies 57 AR genes in S1467 (Table 1). Similar results were observed in ref. 17. Both studies identified catB3, sul1, iscr1, armA, and msrE AR genes in S1467. Authors should delineate the distinctive contributions of this current study from their previous work (ref. 17).
3. Lines 73-76 highlight the aim of this study to comprehensively explore the genetic characteristics of S. Indiana S1467 at the whole-genome level. However, a previous study (ref. 17) already revealed a unique plasmid-free S. Indiana strain, S1467 (ST17), where genomic DNA from 20 S. Indiana strains, including S1467, underwent whole-genome sequencing using the HiSeq 2500 Sequencing platform at Shanghai Biotechnology Corporation (Shanghai, China). Clarification is needed on whether the current study utilized the same sequencing results or generated new sequence data. What distinguishes the outcomes between the two studies?
4. Please include the Blastn results of the genome island analysis as a supplementary figure, referenced in Lines 170-172.
5. Considering this study's identification of 57 AR genes in S1467 based solely on whole-genome sequencing data, PCR validation of specific AR genes such as floR, sul2, tetA (reference PMID: 30157463) is recommended.
6. The Genome island prediction identified a Type IV toxin-antitoxin system (TAS), named YeeVU. TAS comprises two co-expressed genes, one encoding a stable toxin (proteins) and the other encoding unstable antitoxins (proteins or RNA) (Lines 237-238). Conducting Western blot (protein validation, ) or qPCR (RNA validation) is essential to confirm the presence of this system in S1467. This verification significantly enhances the novelty and credibility of this study.
Comments on the Quality of English LanguageMinor editing of English language required
Author Response

(The authors gave the same response as above.)

Round 2
Reviewer 3 Report
Comments and Suggestions for Authors
I don´t understand why the authors "We do not feel very comfortable to submit the raw reads, and are sorry that the accession number is not available".
Reviewer 4 Report
Comments and Suggestions for Authors
Thank you to the author for addressing my concerns and revising the manuscript. I have very few additional comments:
1. The author mentions, "Previous studies identified the AR genes of S1467 using a low-quality draft genome generated from Illumina sequencing results. In this study, AR gene analysis was performed using a high-quality, completed genome derived from the hybrid assembly of Illumina and PacBio sequencing results". Please include relevant details regarding this comparison in the manuscript.
2. The qPCR (RNA validation) conducted to confirm the presence of yeeV in S1467 is commendable. If possible, please consider including this figure in the manuscript for enhanced clarity and completeness.
